# Correlation Analysis of the Quality of Family Functioning and Subjective Quality of Life in Rehabilitation Patients Living with Schizophrenia in the Community

**DOI:** 10.3390/ijerph17072481

**Published:** 2020-04-05

**Authors:** Ling Wang, Xi-Wang Fan, Xu-Dong Zhao, Bing-Gen Zhu, Hong-Yun Qin

**Affiliations:** Department of Psychiatry, Shanghai Pudong New Area Mental Health Center, Tongji University School of Medicine, 165 Sanlin Road, Shanghai 200124, China; wanglinghulixinli@163.com (L.W.); fanxiwang2020@163.com (X.-W.F.); zhaoxd62@gmail.com (X.-D.Z.); binggen.zhu@tongji.edu.cn (B.-G.Z.)

**Keywords:** schizophrenia, rehabilitation in the community, quality of family function, subjective quality of life

## Abstract

*Background:* Recently, the community rehabilitation model for schizophrenia patients has become increasingly popular, and the Shanghai Pudong New Area has developed a relatively complete community rehabilitation model. This study analyzed the correlation between family function and subjective quality of life in the rehabilitation of patients living with schizophrenia in the community. *Methods:* This study evaluated persons living with schizophrenia using the Family Assessment Device and the Subjective Quality of Life Scale. A convenient sampling method was used to select 281 rehabilitation patients living with schizophrenia in the community and 166 hospitalized persons living with schizophrenia. *Results:* There was a significant difference in the Family Assessment Device scores between rehabilitation patients living with schizophrenia in the community and hospitalized persons living with schizophrenia (*p* < 0.0001). The difference in the scores of the subjective quality of life assessment between rehabilitation patients living with schizophrenia in the community and hospitalized persons living with schizophrenia was not statistically significant (*p* > 0.05). The regression analysis showed that quality of family function had a significant effect on the subjective quality of life in rehabilitation patients living with schizophrenia in the community and hospitalized persons living with schizophrenia. (*F* = 10.770 *p* < 0.001), (*F* = 2.960 *p* < 0.01). *Conclusions:* The quality of family function plays an important role in improving the subjective quality of life in rehabilitation patients living with schizophrenia in the community. It may be beneficial to add some methods to improve family function in the current model of rehabilitation in the community.

## 1. Introduction

Persons living with schizophrenia have a chronic mental illness with a high relapse rate and a high disease burden [1]. Although the incidence of schizophrenia is relatively low (median prevalence 15.2 per 100,000 persons per year), it is a high contributor to the total global disease burden [2]. The World Health Organization estimates that the mental illness burden will account for a quarter of the total burden of disease in 2020 [3]. Schizophrenia ranks seventh in the contribution to the nonfatal global burden of disease [4], and persons living with schizophrenia have a disability rate that ranks in the top 10 in the world [5]. Therefore, researchers should focus on this topic.

Various studies have indicated that for persons living with schizophrenia, the family environment is relatively unstimulating and is characterized by a low degree of personality differentiation and rigid thinking [6]. Persons living with schizophrenia have poor family functioning, insufficient family support, and a lack of flexibility in communication, and emotional expression among family members is often extreme [7]. One study reported the negative impact of persons living with schizophrenia on family functioning due to various challenges [8]. Family functioning plays an important role in the development and treatment of schizophrenia [9]. Good family functioning can reduce the recurrence rate and improve the prognosis of schizophrenia [10]. Family functioning has a positive mediating effect on drug compliance in persons living with schizophrenia [11]; therefore, researchers should focus on the family functioning of persons living with schizophrenia.

Chinese persons living with schizophrenia are currently treated mainly based on community rehabilitation methods and hospitalization, and community rehabilitation methods have greatly expanded in recent years [12]. Most persons living with schizophrenia undergo community rehabilitation after symptomatic relief or improvement in the acute phase [13]. In recent years, our research group has devoted itself to the study of rehabilitation of patients living with schizophrenia in the community in Shanghai. The number of hospitalizations and avoidant coping were found to influence the quality of life of people suffering from schizophrenia [14,15].

We found that rehabilitation patients living with schizophrenia in the community rarely go to work and do housework because they live with their family members. It seems that their place of residence has been moved from the ward to the home, i.e., their quality of life is not good enough to achieve the aim of the third-level rehabilitation program in Shanghai. Therefore, we supposed that the current status may be related to family functioning.

## 2. Methods

### 2.1. Design and Procedures

A cross-sectional study was performed. A convenience sampling method was used to select 281 rehabilitation patients living with schizophrenia in the community and 166 hospitalized persons living with schizophrenia. All participants came from the Shanghai Pudong New Area Mental Health Center and its affiliated community rehabilitation institutions. The study lasted from May 2018 to August 2018, and sample collection and scale evaluation were performed by professional clinical researchers, who were responsible for visiting the patients in the community once every 3 months and responding to patient emergencies. The study was supervised by the Psychiatrist of Shanghai Pudong New Area Mental Health Center. The relationship between them was similar to that between a family doctor and resident. During the recruitment process, the patients were asked to complete the questionnaires on a certain day.

### 2.2. Study Population

The sample of persons living with schizophrenia was originally recruited from the Shanghai Pudong New Area Mental Health Center and its affiliated community rehabilitation institutions. This study had been reviewed by the Shanghai Pudong New Area Mental Health Center (Shanghai, China) with the approval of the Review Board, in accordance with the principles of the Declaration of Helsinki. Ethics approval was granted by the Ethics Committee of Shanghai Pudong New Area Mental Health Center.

The inclusion criteria were as follows: patients who (1) met the diagnostic criteria for schizophrenia in accordance with the 10th edition of the Statistical Classification of International Diseases and Related Health Problems; (2) were actively taking atypical antipsychotics; (3) were between the ages of 18 and 65 years; (4) had no history of organic brain disease or head trauma, obvious intellectual disability, or other serious or uncontrolled stable physical illnesses; (5) had no current (within 3 months) alcohol or drug use or a history of past dependence; and (6) had no obvious complaints of hallucinations or delusions.

## 3. Measures

### 3.1. Family Assessment Device

The Family Assessment Device is a self-reported scale designed to measure family members’ perceptions of family functioning [16]. The Family Assessment Device includes seven dimensions, including six subscales. The seven dimensions are (1) problem solving (PS). When maintaining an effective level of family function, the family’s ability to solve problems refers to the ability to solve the problems that threaten the integrity and functional capacity of the family; (2) communication (CM) refers to the information exchange of family members, focusing on whether the content of the verbal information is clear and whether the information is directly transmitted; (3) roles (RL) refers to the family and whether a behavioral model for completing a series of family functions is established, whether the task division is clear and fair, and whether family members have completed the task seriously; (4) affective responsiveness (AR) assesses the extent of family members’ emotional response to stimuli; (5) affective involvement (AI) assesses the degree to which family members care and value each other’s activities and possessions; (6) behavior control (BC) assesses whether a family has different behavioral control modes in different situations; (7) general functioning (GF) refers to an overall evaluation of family functions [17]. The range of family functions measured by this scale makes it suitable for clinical application, and the scale has good reliability and validity [18]. The scores of the subscale are the total scores of each item which ranges from 1 to 4. If 40% of the items on a subscale are not answered, the subscale will not be scored [19]. The lower the score, the less healthy the family structure and functioning [20]. The Family Assessment Device has been widely used in various research environments [21]. In most studies, it has been able to distinguish between clinical populations and controls as well as patient populations with different diseases [21]. Some studies have found that the Family Assessment Device has good adaptability and sensitivity in populations in mainland China [22].

### 3.2. Schizophrenia Quality of Life Scale

The Schizophrenia Quality of Life Scale was developed by the British psychiatrist Greg Wilkinson, and additional revised self-reporting scales have been created by Luo Hong and others [23,24]. The questionnaire items are rated on a 5-point Likert-type scale. The Schizophrenia Quality of Life Scale includes three subscales: psychosocial subscale, reflects the content composition of emotional expression and interpersonal communication; motivation/energy subscale, reflects the content composition of motivation and energy; and symptom and adverse reaction subscale, reflects the content composition of symptoms and side effects of drugs [25]. The total raw score ranges from 0 to 120, the lower the score, the better the subjective quality of life [26]. Previous studies have confirmed that this subjective quality of life scale is an effective and reliable tool for testing the quality of life in persons living with schizophrenia [27].

### 3.3. Statistical Analysis

All results are expressed as the mean ± standard deviation. Statistical analysis of the data was performed using the SPSS 22.0 software package (SPSS, Chicago, IL, USA). The chi-square test was conducted to compare the general characteristics of the study participants. Independent-samples *t*-tests for family functioning and subjective quality of life, linear regression analysis for the correlation between the two factors, and Pearson’s correlation analysis were used to examine the relationships between the variables. The associations between variables were considered statistically significant at a level of *p* < 0.05.

## 4. Results

### 4.1. Demographic Data of All the Participants Enrolled in This Study

The descriptive and statistical comparisons of community and hospitalized persons living with schizophrenia are shown in Table 1. There were no significant differences in the sociodemographic data between the two groups.

### 4.2. Analysis of the Differences in Family Functioning between Community and Hospitalized Persons Living with Schizophrenia

The Family Assessment Device scores of rehabilitation patients living with schizophrenia in the community were significantly lower than those of hospitalized persons living with schizophrenia (*p* < 0.01); in addition, the subscale scores were significantly different between the two groups (*p* < 0.01). See Table 2 for the more details.

### 4.3. Analysis of Subjective Quality of Life in Community and Hospitalized Persons Living with Schizophrenia

There was no significant difference in the subjective quality of life scores between the community and hospitalized persons living with schizophrenia. The difference in the motivation and energy subscale scores between the two groups of patients was statistically significant (*p* < 0.05), and the results showed that community rehabilitation failed to improve the quality of life of patients among persons living with schizophrenia. See Table 3 for more details.

### 4.4. Linear Regression Analysis of Family Functioning and Subjective Quality of Life in Rehabilitation Patients Living with Schizophrenia in the Community and Hospitalized Persons Living with Schizophrenia

The regression analysis of quality of family functioning and subjective quality of life showed that family functioning had a significant effect on the subjective quality of life in rehabilitation patients living with schizophrenia in the community (*F* = 10.770 *p* < 0.001). See Table 4 for more details.

The regression analysis of quality of family functioning and subjective quality of life showed that family functioning had a significant effect on the subjective quality of life in hospitalized persons living with schizophrenia (*F* = 2.960 *p* < 0.01). See Table 5 for more details.

In summary, the results of regression analysis showed that compared with hospitalized persons living with schizophrenia, the problem solving of family function had a greater impact on the subjective quality of life of rehabilitation patients living with schizophrenia in the community.

## 5. Discussion

There have been many studies on family functioning and quality of life in persons living with schizophrenia [28]. The current research focused on family functioning and subjective quality of life in community rehabilitation and hospitalized persons living with schizophrenia [29]. The results showed that the family functioning scores of rehabilitation patients living with schizophrenia in the community were lower than those of hospitalized patients, indicating that rehabilitation patients living with schizophrenia in the community have better family functioning than that of hospitalized persons living with schizophrenia. These findings are similar to those of previous studies [30]. Previous studies have found that the family functioning of persons living with schizophrenia was significantly different from that of ordinary families and was characterized by an unstimulating family atmosphere, rigid thinking, and lack of emotional expression [31]. Persons living with schizophrenia undergoing community rehabilitation who are living with their parents are able to feel the love and support of their loved ones [32]. Some studies have shown that parental support is related to family functioning [33], and hospitalized persons living with schizophrenia are less likely to feel love from their parents.

There was a small but significant difference in quality of life between the community and hospitalized groups, indicating that community rehabilitation did not effectively improve patients’ quality of life. Previous research has shown that even though Shanghai has a good community rehabilitation medical policy, these policies do not effectively improve the quality of life of rehabilitation patients living with schizophrenia in the community [34]. Although patients with mental illness who are in community rehabilitation programs are stable and have high drug compliance, their perceived illness and quality of life are poor, and their perceived illness is greatly impacted by family members [35]. The quality of life of persons living with schizophrenia is related to a variety of factors, such as drug side effects, social support, and stigma [36]. Previous studies have also found a negative correlation between quality of life and stigma in persons living with schizophrenia [37]. Studies have also found that the quality of life of persons living with schizophrenia is significantly better than that of outpatients [38]. In this study, there was almost no difference in subjective quality of life between community and hospitalized patients, which may be partly due to effective hospital-based mental health rehabilitation programs in Pudong New Area, Shanghai. Hospitalized persons living with schizophrenia have the opportunity to participate in good rehabilitation programs, with activities such as calligraphy competitions and charity bazaars. In addition, hospitalized patients often receive positive social support from medical staff and other patients [39]. Some studies have found that social support and quality of life are positively correlated [40].

### Limitations

The sample source of this study was a single center, and the results are considered preliminary. This study did not explore other factors that could have a significant impact on the quality of life and family functioning of people who suffer from schizophrenia [41].

## 6. Conclusions

The study found that the quality of family functioning of rehabilitation patients living with schizophrenia in the community is much better than that of hospitalized persons living with schizophrenia, but their subjective quality of life has not improved significantly. There was a highly positive correlation between rehabilitation patients living with schizophrenia in the community and quality of family functioning and subjective quality of life. Perhaps some methods can be applied to improve family functioning, as well as enrich their quality of life, in the current model of rehabilitation in the community.

## Figures and Tables

**Table 1 ijerph-17-02481-t001:** General characteristics of the study participants.

Item	Total		RSC		HPS		$x^{2}$
	*N* = 447%		*N* = 281%		*N* = 166%		
Age, years							3.21
≤30	24	5.37	13	4.60	11	6.62	
31 to 45	206	46.08	131	46.60	75	45.19	
46 to 60	217	48.55	137	48.80	80	48.19	
Gender							0.011
Male	242	54.14	145	51.60	97	58.43	
Female	205	45.86	136	48.40	69	41.57	
Education level							6.568
Elementary school	21	4.78	16	5.69	5	3.01	
Junior high school	171	38.76	117	41.64	54	32.53	
Senior high school	182	40.73	115	40.93	67	40.36	
University education	70	15.66	31	11.03	39	23.49	
Graduate education	3	0.07	2	0.71	1	0.61	

Notes: the analysis method used was the chi-square test. RSC, rehabilitation patients living with schizophrenia in the community; HPS, hospitalized persons living with schizophrenia.

**Table 2 ijerph-17-02481-t002:** Family functioning analysis results of community and hospitalized persons living with schizophrenia.

	RSC	HPS	*p*
Problem Solving	12.64 ± 2.41	13.54 ± 2.41	**0.0001**
Communication	19.84 ± 3.31	21.05 ± 2.61	**0.0001**
Roles	25.37 ± 3.55	26.75 ± 3.81	**0.0001**
Affective Responsiveness	14.69 ± 2.39	13.91 ± 2.67	**0.002**
Affective Involvement	16.49 ± 2.72	17.95 ± 3.23	**0.0001**
Behavior Control	19.82 ± 2.83	21.23 ± 2.90	**0.0001**
General Functioning	24.71 ± 4.64	27.34 ± 4.23	**0.0001**

Notes: the data are presented as the mean ± SD. The analysis method used was an independent samples *t*-test. The bold values are statistically significant. RSC, rehabilitation patients living with schizophrenia in the community; HPS, hospitalized persons living with schizophrenia.

**Table 3 ijerph-17-02481-t003:** Subjective quality of life of community and hospitalized persons living with schizophrenia.

	RSC	HPS	*p*
Total score	25.23 ± 14.59	25.86 ± 15.23	0.668
Psychosocial	10.88 ± 9.40	11.98 ± 8.92	0.225
Motivation and energy	11.31 ± 4.51	10.22 ± 4.88	**0.017**
Symptoms and side effects	3.03 ± 3.96	3.64 ± 4.32	0.128

Notes: the data are presented as the mean ± SD. The analysis method used was an independent samples *t*-test. The bold values are statistically significant. RSC, rehabilitation patients living with schizophrenia in the community; HPS, hospitalized persons living with schizophrenia.

**Table 4 ijerph-17-02481-t004:** Rehabilitation patients living with schizophrenia in the community quality of family functioning and subjective quality of life regression analysis results.

		NSC		NC	*t*	*p*	*VIF*	*R* ^2^	Amend *R*^2^		*F*
		B	Standard Error	Beta							
Constant	−26.064	7.232	-	−3.604	0.000 **	-	0.216	0.196		*F* (7, 273) = 10.770, *p* = 0.000	
PS	1.735	0.546	0.287	3.177	0.002 **	2.841					
CM	0.409	0.386	0.093	1.060	0.290	2.681					
RL	0.529	0.316	0.129	1.671	0.096	2.066					
AR	−1.042	0.475	−0.191	−2.195	0.029 *	2.639					
AI	0.328	0.393	0.061	0.836	0.404	1.877					
BC	0.631	0.358	0.122	1.762	0.079	1.684					
GF	0.177	0.351	0.056	0.505	0.614	4.346					
Dependent variable: SQLS											
D–W value: 1.937											

Notes: * correlation is significant at a confidence level of 0.05; ** correlation is significant at a confidence level of 0.001. The analysis method used was regression analysis. NSC, nonstandardized coefficient; NC, normalization coefficient, PS, problem solving; CM, communication; RO, roles; AR, affective responsiveness; AI, affective involvement; BC, behavior control; GF, general functioning; SQLS, Schizophrenia Quality of Life Scale.

**Table 5 ijerph-17-02481-t005:** Hospitalized persons living with schizophrenia quality of family functioning and subjective quality of life regression analysis results.

		NSC		NC	*t*	*p*	*VIF*	*R* ^2^	Amend *R*^2^		*F*
		B	Standard Error	Beta							
Constant	6.636	11.876	-	0.559	0.577	-	0.116	0.077		*F* (7, 158) = 2.960, *p* = 0.006	
PS	1.504	0.678	0.238	2.219	0.028 *	2.060					
CM	−0.142	0.620	−0.024	−0.229	0.819	2.014					
RL	0.160	0.455	0.040	0.351	0.726	2.321					
AR	−0.263	0.655	−0.041	−0.401	0.689	1.894					
AI	−0.497	0.514	−0.106	−0.968	0.335	2.126					
BC	−0.238	0.553	−0.045	−0.431	0.667	1.989					
GF	0.564	0.458	0.157	1.232	0.220	2.895					
Dependent variable: SQLS											
D-W value: 1.812											

Notes: * correlation is significant at a confidence level of 0.05. The analysis method used was regression analysis. NSC, nonstandardized coefficient; NC, normalization coefficient, PS, problem solving; CM, communication; RO, roles; AR, affective responsiveness; AI, affective involvement; BC, behavior control; GF, general functioning; SQLS, Schizophrenia Quality of Life Scale.

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
