# Peer review of "Correlation Analysis of the Quality of Family Functioning and Subjective Quality of Life in Rehabilitation Patients Living with Schizophrenia in the Community"

_ijerph, 2020, doi:10.3390/ijerph17072481_

Round 1

Reviewer 1 Report

This paper addresses an interesting angle on the health and wellbeing of people with schizophrenia, i.e. the relationship between family functioning and QOL in community and hospitalised schizophrenia patients.

The introduction is fine but there is scope to link the broad aims of the study with an understanding about why this might be important in terms of community management of these patients. Is the point here that by better understanding the link between family functioning and QOL that interventions might be developed to support families and thereby improve QOL and clinical outcomes in community based schizophrenia patients?

The final paragraph of the introduction states the aims but these are rather diffuse and could do with broader explanation. At the moment there are three aims: a comparative study about family functioning and QOL in community and hospitalised patients, the efficacy of rehabilitation, and the correlation between family functioning and QOL in all schizophrenia patients.

I am not sure the study and the methods used are able to address all these questions, or at least the current description of the sample and the methods doesn't support all three aims.

Specifically:

1) It is not clear how the sample was recruited. Was this a consecutive or convenience sample? 

2) All the patients have been hospitalised but at what point did some of them become community based - these is no distinction made in the recruitment or timing of recruitment about how the sample was split between the two groups.

3) What does sample evaluation mean - perhaps this relates to point two above.

4) Who is the professional clinical researcher and what is their relationship to the sample and how it was recruited?

5) The linear regression is undertaken on the total sample and so again there seems to be a mismatch between the aims and the use of the sample as there is no distinction here between community and hospital based.

6) In the absence of controlled data with two time points this study cannot say anything about the efficacy of the rehabilitation programmes and I would suggest the authors down play any claims about this finding.

Reviewer 2 Report

I am a pleasure to review your manuscript. I think this is a significant topic. Your findings showed that the failure to improve the subjective quality of life in community rehabilitation patients with schizophrenia was partly due to family dysfunction.

Unfortunately, I cannot understand your conclusion because of a lack of detailed explanations of why this analysis was performed. Additionally, I think your study can be better to change the analysis model from a univariate to a multivariate analysis model. Some bias could be influential to your findings of univariate analysis. I hope for your further consideration of the changing analysis model.

[Major points]
L61.I think it is required to mention the ethical consideration for research subjects. If your study has been reviewed by an ethical committee, you should mention it in your manuscript.

L79,89. You need a detail explanation about the subgroups in the two scales (Family Assessment Device and Schizophrenia QOL). I could not understand your findings because of a lack of information about each subgroup.

L97. In this study, you have implemented a univariate analysis such as the correlation, however, the evidence obtained findings are low level. I think it is a critical limitation in which the bias has not been removed.

L130. Please show the subjects that you have implemented the linear regression analysis. If not wrong, the subjects of this analysis were the only community rehabilitation patients(excluded hospitalized patients)?

[Minor points]
L142. In the correlation analysis table, it is necessary to show statistically significant results while using * or bold texts.

Reviewer 3 Report

Thank you for the opportunity to review this manuscript. The authors examined to compare family functioning and quality of life in community and hospitalized patients with schizophrenia and evaluated the community-based rehabilitation of patients with schizophrenia using a sample from a single centre in Shanghai Pudong New Area.

First, the theme they addressed was interesting and important for practitioners, researchers, and/or service users. But, the methods of data analysis employed are inaccurate for the aim of this study. In other words, the results obtained from the data analysis are also questionable. For example, the authors performed only simple statistics, and t-tests and regression analysis did not consider the effects of confounders. These results of the reanalysis may change, and the authors should reconsider based on the results. Second, there was not enough information for readers to understand the main objective variables, Family assessment device and Schizophrenia QOL. For the Family assessment device, if the total score is the total value of each subscale including problem-solving, communication, roles, affective responsiveness, affective involvement, and behaviour control, an incorrect value was described. Third, I could not find the statement about the ethical approval for this research in this manuscript.

I hope my review comments will help improve this research paper.

Round 2

Reviewer 2 Report

I am pleased to have an opportunity to review your manuscript. I can understood it is useful for mental disabilities in comunities that to be taken rehabilitation services can be high QOL. However, I think there are several points to be modified in your present manuscript.

[Critical points]
I think it is necessary to mention the ethical consideration for research subjects. If your study has been reviewed by an ethical committee, you should mention it in your manuscript.

[Major points]
L160: You implemented liner regression analysis between family functioning and subjective quality of life "in patients with schizophrenia". Thanks to your analysis represented Table 4.I can understand the association between family function and subjective QOL. How about the association between family functioning and subjective quality of life "in hospitalized person with schisoprenia"? I think you can evaluate whether rehabilitation in the community can contribute to their QOL if you compare the βvalue between in community or in hospital patients.

L209: You mentioned that "The study found that the quality of family functioning of rehabilitation patients living with schizophrenia in the community is much better than that of hospitalized persons living with schizophrenia". However, if you would like to conclude as like above, it needs an additional regression analysis between quality of family functioning and subjective quality of life in HOSPITALIZED patient with schizophrenia.

[Minor points]
L100: It was a minor mistake. You skipped no.⑤.

L164: Please delete "See Figure 1 for more details." I think that figure 1 has already removed.

Reviewer 3 Report

Thank you for the opportunity to review this manuscript.

First, your response to my suggestions is insufficient, especially in the data analysis. You stated that you re-analyzed, but the actual results have not changed. My suggestion is that you should performed a multivariate analysis for considering confounders. This is a critical limitation in which the bias has not been removed. I hope for your further consideration of the changing analysis model.

Second, in Family assessment device as one of the main objective variable, you should show the range of both each subscale and total score.

It would be helpful if you could write the Response letter carefully. I couldn't figure out what part, why, and how you fixed it.

I hope my review comments will help improve this research paper.
